# High-Density Lipoprotein Modifications: A Pathological Consequence or Cause of Disease Progression?

**DOI:** 10.3390/biomedicines8120549

**Published:** 2020-11-28

**Authors:** Andrea Bonnin Márquez, Sumra Nazir, Emiel P.C. van der Vorst

**Affiliations:** 1Institute for Molecular Cardiovascular Research (IMCAR), RWTH Aachen University, 52074 Aachen, Germany; anbonninmarq@ukaachen.de (A.B.M.); snazir@ukaachen.de (S.N.); 2Interdisciplinary Center for Clinical Research (IZKF), RWTH Aachen University, 52074 Aachen, Germany; 3Department of Pathology, Cardiovascular Research Institute Maastricht (CARIM), Maastricht University Medical Centre, 6229 ER Maastricht, The Netherlands; 4Institute for Cardiovascular Prevention (IPEK), Ludwig-Maximilians-University Munich, 80336 Munich, Germany; 5German Centre for Cardiovascular Research (DZHK), partner site Munich Heart Alliance, 80336 Munich, Germany

**Keywords:** high-density lipoproteins, inflammation, HDL modifications, dysfunctional HDL

## Abstract

High-density lipoprotein (HDL) is well-known for its cardioprotective effects, as it possesses anti-inflammatory, anti-oxidative, anti-thrombotic, and cytoprotective properties. Traditionally, studies and therapeutic approaches have focused on raising HDL cholesterol levels. Recently, it became evident that, not HDL cholesterol, but HDL composition and functionality, is probably a more fruitful target. In disorders, such as chronic kidney disease or cardiovascular diseases, it has been observed that HDL is modified and becomes dysfunctional. There are different modification that can occur, such as serum amyloid, an enrichment and oxidation, carbamylation, and glycation of key proteins. Additionally, the composition of HDL can be affected by changes to enzymes such as cholesterol ester transfer protein (CETP), lecithin-cholesterol acyltransferase (LCAT), and phospholipid transfer protein (PLTP) or by modification to other important components. This review will highlight some main modifications to HDL and discuss whether these modifications are purely a consequential result of pathology or are actually involved in the pathology itself and have a causal role. Therefore, HDL composition may present a molecular target for the amelioration of certain diseases, but more information is needed to determine to what extent HDL modifications play a causal role in disease development.

## 1. Introduction

Lipoproteins are a broad class of macromolecules, which are composed of cholesterol, triglycerides, phospholipids, and various other proteins. These molecules are most commonly classed into five main sub-categories based on their buoyant density: chylomicrons, very-low-density lipoprotein (VLDL), low-density lipoprotein (LDL), intermediate-density lipoprotein (IDL), and the smallest and most dense of the five is high-density lipoprotein (HDL) [1]. Many studies over the last decades focused on HDL-cholesterol (HDL-C), commonly known as the “good cholesterol”, which was found to have an inverse relationship with cardiovascular disease (CVD). In a meta-analysis of four large epidemiological studies from the 1970s and 1980s, Gordon et al. found that a 1 mg/dL increase in HDL-C resulted in a 2–3% reduction in risk for coronary heart disease (CHD) [2]. This relationship can be attributed to the observation that HDL ameliorates atherosclerosis [3], which is recognized as the main cause of CVD [4]. This apparent cardiovascular-protective property of HDL-C gave rise to the HDL hypothesis, stating that increasing HDL-C levels would result in lower incidences of cardiovascular events, which fueled the interest in its therapeutic potential [5,6]. HDL plays a crucial role in the reverse cholesterol transport (RCT) and, thus, in the efflux of cholesterol from the vasculature [7]. Additionally, HDL exhibits anti-oxidative, anti-thrombotic, and anti-inflammatory properties [8,9], which altogether contribute to its cardiovascular-protective effects. However, a wide range of clinical and genetic studies have demonstrated that HDL-C is only a limited indicator of CVD risk and, thereby, not the most appropriate therapeutic target [10,11]. It is becoming more and more evident that HDL function is the more important determinant for disease outcome, compared to HDL-C [12]. Multiple studies have already shown that HDL in patients suffering from a wide range of pathologies, such as diabetes mellitus, rheumatoid arthritis (RA), renal diseases, and cancer can cause modifications in HDL leading to impaired HDL function and the manifestation of adverse effects [13,14,15]. Modifications of HDL can occur due to general inflammatory processes [16], but also due to disease-specific processes, such as carboxylation, oxidation, or glycation [12]. In addition, genetic factors can also cause modifications of the HDL particles [17]. Therefore, it is highly interesting to uncover which modifications can occur in the HDL particles and how these affect HDL function and disease outcome. This review will highlight the complex HDL molecule, the potentially detrimental effects that modifications to the main proteins and the overall composition can have and the difficulties to determine whether these modifications are only a consequence of the pathology, or can also play a causal role in disease progression. Furthermore, some strategies that are currently being investigated to enhance HDL functionality or to reverse the noxious modifications will be discussed.

### 1.1. HDL Structure and Metabolism

HDL, the smallest and most dense lipoprotein, is composed of a central hydrophobic non-polar lipid core, consisting of primarily triacylglycerols and cholesterol esters. This hydrophobic core is surrounded by a surface monolayer, ~20 Å thick, comprising of apoproteins, phospholipids, and non-esterified cholesterol [18]. In human plasma, HDL is comprised of a highly heterogeneous population of particles ranging in size from 5 to 17 nm and with a density between 1.063–1.210 g/mL [19,20]. Using proteomic and lipidomic techniques, it has been demonstrated that more than 200 different lipids as well as over 85 proteins can be present in HDL particles [19]. This accounts for the large variability in HDL particles which can be fractionated and divided into various subtypes by different techniques according to their physicochemical properties as summarized in Table 1. Using analytic ultracentrifugation, a gold standard technique for HDL separation, human HDL can be separated into two subclasses based on density, the less dense HDL_2_ (1.063–1.125 g/mL) and more dense HDL_3_ (1.125–1.21 g/mL) [21]. HDL_2_ and HDL_3_ can be further sub-classified on the basis of their particle size by gradient gel electrophoresis into two HDL_2_ (HDL_2_b(10.6 nm), HDL_2_a(9.2 nm)) and three HDL_3_ (HDL_3_a(8.4 nm), HDL_3_b(8.0 nm), and HDL_3_c (7.6 nm)) subclasses [22]. On the basis of electrophoretic mobility, HDL can also be subdivided into two main subpopulations: α-HDL and pre-β HDL [23,24]. Moreover, using immunoaffinity methods, HDL can be separated into particles containing apolipoprotein A1 (apoA-1) with or without apoA-2 (LpA-1:A-2 or LpA-1, respectively) based on its apolipoprotein composition [25]. The major protein component of HDL is apoA-1, which is primarily synthesized in the liver (~80%) and intestine (~20%) and is secreted in a lipid-free state. Secreted apoA-1 interacts with the transporter molecule ATP-binding cassette transporter A1 (ABCA1), which facilitates the transfer of cholesterol and phospholipids to lipid-poor apoA-1 as shown in Figure 1. Through a series of intermediate steps, lipid-free apoA-1 is gradually converted into a pre-β-migrating or nascent discoidal HDL particle enriched in unesterified cholesterol [26]. Enzymes such as lecithin-cholesterol acyltransferase (LCAT), cholesterol ester transfer protein (CETP), and phospholipid transfer protein (PLTP) are subsequently incorporated and are responsible for the lipid and phospholipid composition of nascent discoidal HDL. Subsequent lipidation by these enzymes leads to the generation of mature spherical HDL, which can vary in size, density, and composition [27]. Incorporation of various proteins and lipids, which all interact to regulate the functional properties of HDL, result in further refinement of HDL particles.

### 1.2. HDL Function and Remodeling

HDL has been postulated to exert numerous important functions that may contribute to the protection from atherosclerosis and other inflammatory diseases as previously mentioned. The best documented of these functions is the ability of HDL to promote cholesterol efflux from cells through RCT reducing atherosclerotic plaque sizes, by which it also plays an important role in lipid homeostasis [28]. HDL also exhibits an antioxidant capacity, as especially HDL_3_ has been shown to protect LDL and other lipoproteins from free-radical induced oxidative damage and, thereby, inhibit the generation of pro-inflammatory oxidized lipids [29,30]. Additionally, HDL is an anti-inflammatory particle, although in the presence of acute or chronic inflammation it may become pro-inflammatory as will be discussed later [31]. HDL is a complex molecule that can be modified due to pathological conditions affecting its functionality, which in turn is associated with disease progression. 

As can be expected, remodeling of HDL, either by changes in the lipidome or proteome or by modifications to different components, can have a severe impact on the functionality of HDL [17,32]. Even within the healthy population the different subtypes of HDL, which vary in their composition, have different functions and can exert different protective roles. Thus, HDL which normally exerts beneficial effects can paradoxically become a pro-inflammatory, atherogenic molecule, which plays a detrimental role in various pathologies [33]. Therefore, the effects of various pathological states on the composition and thereby function of HDL will be discussed in more detail, primarily focusing on the resulting modulation of proteins and changes to the composition and structure of HDL. Identification of these modifications and especially elucidation of the functional and causal consequences will provide important insights to develop novel therapeutic approaches focusing on HDL composition and functionality. 

## 2. Post-Translational Modifications of HDL

Post-translational modifications (PTMs) of proteins have gained interest in the field of CVD, as proteins are being frequently exposed to different tissue components and plasma under diverse pathophysiological conditions. Here, we will highlight the most commonly reported PTMs, such as oxidation, carbamylation, and glycation.

### 2.1. Oxidation

Oxidation is one of the most important processes involved in the pathogenesis of diseases. HDL oxidation takes place predominantly in inflammatory microenvironments [34]. Myeloperoxidase (MPO) and superoxide anion radical (O_2_^−^) are secreted by active phagocytes, that thereby might be potential candidates responsible for the generation of oxidized/modified lipoproteins [35]. Zheng et al. showed the clinical relevance of such modifications and demonstrated that apoA-I is a target of MPO-catalyzed modifications, such as chlorination and nitration in serum of cardiovascular disease subjects [36]. These modifications are associated with the loss of lipid binding properties of apoA-I and also impairs ABCA1 dependent efflux of cholesterol from macrophages [36]. Furthermore, two sulfur-containing residues (cysteine and methionine) are normally found in proteins. Oxidation of methionine residues in apoA-I, C-II and C-III of HDL has been shown to induce changes in the activity of these proteins [37,38]. For example, sulfur oxidation of methionine generates pro-inflammatory HDL, as it induces the accumulation of tumor necrosis factor alpha (TNF-α) and interleukin-6 (IL-6) in human primary monocytes and mouse bone marrow-derived macrophages [39].

### 2.2. Carbamylation

Carbamylation is a non-enzymatic and irreversible PTM that results from the interaction between isocyanic acid and various amino groups of proteins. Protein carbamylation takes place in an inflammatory environment as a result of MPO-dependent cyanate formation. It has been reported that carbamylation of HDL results in the loss of anti-oxidative and anti-inflammatory properties of HDL. Carbamylation also suppresses the activity of LCAT, which leads to the generation of dysfunctional HDL [40]. Numerous studies have shown that carbamylation of apoA-I is associated with CVD, as elevated levels of carbamylated apoA-I can be observed in plaques of human atherosclerotic subjects compared to healthy controls [41]. Therefore, carbamylated proteins, such as carbamylated apoA-I, might be an independent risk factor for cardiovascular mortality and morbidity.

### 2.3. Glycation and Advanced Glycation End Products

Advanced glycation end products (AGEs) are lipids or proteins that undergo glycation after exposure to sugars. These compounds not only cause detrimental effects in diabetic patients, but have also been associated with the development of cardiovascular complications, such as hypertension, endothelial dysfunction, diastolic function, increased myocardial and vascular stiffness, and formation of atherosclerotic lesions [42]. Glycation of apoA-I adversely affects the enzymatic activity and biological functions of HDL such as LCAT, CETP, and paraoxonase (PON) [43,44]. For example, methylglyoxal-dependent modifications of lysine, arginine, and tryptophan residues altered the conformation of an apoA-I epitope in regions critical for the binding of lipids and LCAT activation [43]. Furthermore, it has been shown that non-enzymatic glycation of apoA-I impairs the anti-inflammatory properties of HDL and decreases neutrophil infiltration into media/intima of carotid arteries [45]. Recent studies have demonstrated that apoA-I glycation decreases its stability and results in a threefold shorter half-life of apoA-I, leading to the generation of dysfunctional HDL in type 2 diabetes [46].

## 3. HDL Modifications

Modifications to HDL can occur in many different pathologies, thereby affecting the lipidome and proteome and, thus, functionality of HDL. This section will review some of the key actors that play a role in the pathological modification of HDL and thereby compromise the functionality of HDL. Furthermore, the question of whether such modifications are only a pathological consequence or can also act as a causal factor in disease development will be discussed. 

### 3.1. Serum Amyloid A

Serum amyloid A (SAA) is a family of apolipoproteins that are primarily synthesized in the liver and that play a pivotal role in innate immunity [47] as one of the main proteins of the acute phase response [48]. SAA synthesis is induced by acute and chronic inflammation as evidenced by increased levels of SAA in patients with acute inflammation or with chronic inflammatory disorders, such as RA, obesity or type 2 diabetes [49]. Moreover, evidence has shown that SAA is involved in the lipid metabolism and transport [50] and is critical in the regulation of inflammation [47]. SAA can replace important proteins in the HDL particle resulting in compositional changes (Figure 2) and loss of its anti-inflammatory and atheroprotective potential [51]. With increasing serum levels during inflammation, the levels of SAA associated with HDL also increase and studies further link higher levels of SAA to increased incidences of CVD and atherosclerosis [52]. For these reasons, SAA modifications of HDL are extensively studied and its role will be shortly summarized in this section. 

Relative to apoA-1, the binding affinity of SAA is much greater, allowing SAA to completely displace this key protein [53]. This increased affinity may be due to SAA having a lipophilic surface as opposed to apoA-1, which has a hydrophobic interior, which must be opened and exposed in order to bind to lipids, additionally, the radius of surface curvature of SAA is quite similar to that of HDL, thus, allowing for a neat fit [54]. The ability of SAA to replace apoA-1 has a significant impact on HDL composition and on the functionality of HDL [48,49,55]. This displacement of apoA-1 by SAA is evident in HDL of patients with CHD, end-stage renal disease (ESRD), and sepsis, all of which display increased levels of SAA and decreased levels of apoA-1 in HDL in comparison to healthy controls [55,56,57]. Since apoA-1 plays a major role in RCT, its displacement and subsequent incorporation of SAA to HDL compromises the ability of HDL to mediate cholesterol efflux from macrophages [58,59], inherently limiting the anti-atherogenic potential of HDL. In addition to apoA-1, SAA is also capable of displacing other proteins, such as PON1 and platelet-activating factor acyl hydrolase (PAF-AH), which are important mediators of the antioxidant properties of HDL [32]. Zewinger et al. showed evidence, both in vitro and in vivo, that SAA can turn HDL from a protective lipoprotein into one that is harmful [60]. Their examination of a large cohort of patients demonstrated an increased all-cause mortality in association to high SAA, indicating that SAA accumulation results in dysfunctional HDL [60]. Moreover, SAA has been shown to be present in atherosclerotic lesions in coronary and aortic arteries in mice [61] and humans [62,63]. In line with this, it was found that SAA enrichment of HDL in ESRD leads to the loss of the anti-inflammatory properties of HDL and increased proatherogenic effects [64], showing not only a loss of function due to changes in composition but a complete reversal in the pathophysiological effects of HDL. Similar effects have been found in patients with type-2 diabetes, in which the anti-inflammatory properties of HDL are also impaired due to increased SAA concentrations [65,66,67]. Although it has been generally shown that SAA enriched HDL has detrimental effects, SAA bound to HDL may also play a role in downregulating some inflammatory effects. For example, the incorporation of serum SAA into HDL reduces the SAA levels in serum, thus, reducing the pro-inflammatory effects it exerts as an independent protein [68]. Serum SAA can activate the NLR family pyrin domain containing 3 (NLRP3) inflammasome and initiate the downstream secretion of pro-inflammatory cytokines [69], thus integration into HDL molecules can result in an impaired capacity to activate NLRP3 [68]. In vitro studies on THP-1 cells, demonstrated that the addition of HDL reduces the expression of NLRP3 and its downstream inflammatory cytokines. However, treatment of cultures with modified HDL, in the form of oxidized HDL, not only increases the expression of NLRP3, but can also activate downstream inflammatory cytokines [70].

Overall, the evidence clearly delineates the damaging effects associated with SAA enriched HDL. SAA can directly replace the major beneficial proteins in HDL, thus, abolishing the beneficial properties they confer, and as a mediator of inflammation, SAA can also indirectly contribute to the modification of various components in HDL. The above described studies highlight that HDL-associated SAA plays a significant role in the modification of HDL in various pathologies. However, reduction of SAA serum levels by HDL binding can also confer some anti-inflammatory effects. SAA is one of the major contributors to changes in HDL composition and to its transformation into dysfunctional HDL. However, most of the above described studies only demonstrate that SAA incorporation into HDL is a consequence of the pathological condition. Although some initial studies demonstrate that HDL modified by SAA has also causal and adverse effects, more studies are still required to elucidate the exact causal role of dysfunctional SAA-enriched HDL in disease development.

### 3.2. Lecithin-Cholesterol Acyltransferase

LCAT is a crucial enzyme in the lipoprotein metabolism as it esterifies free cholesterol and phosphatidylcholine into cholesteryl esters and allows for the maturation of the HDL particle. LCAT is predominantly activated by apoA-1 but can also be activated, though less effectively, by various other apolipoproteins [71]. The capacity of LCAT to esterify free cholesterol gives it an important role in the maintenance of cholesterol homeostasis and in the regulation of the cholesterol transport and thus the RCT [72]. Inflammation has been shown to negatively affect LCAT, as pro-inflammatory cytokines like TNF-α and IL-6 appear to diminish its activity [73,74]. Additionally, the increased MPO levels that are present during inflammation increase MPO-mediated HDL oxidation and decrease LCAT activation adversely affecting RCT [75,76,77]). Decreased LCAT activity has also been observed during the acute phase response [78]. LCAT inactivity results in modifications to HDL since LCAT is responsible for transforming discoidal nascent HDL into mature spherical shaped HDL. Additionally, the esterification of cholesterol allows the lipid to preferentially bind to the HDL particle, thus, affecting its composition [79]. This binding of cholesteryl esters not only changes composition but is the basic tenet behind the anti-atherogenic potential of HDL. This evidence shows that an inflammatory environment eventually leads to loss of LCAT functionality, which directly affects the functionality of HDL. Gebhard et al. conducted a study in coronary artery disease (CAD) patients and found that an upregulation of LCAT protein concentration in plasma was accompanied with lower plaque volumes, confirming the atheroprotective effects of LCAT [80]. In RA and sepsis patients, LCAT activity is also significantly decreased when compared to healthy controls. Decreased activity was also found to be associated with lower HDL levels [81,82]. Dysfunctionality of HDL in these inflammatory pathologies has a cyclic effect that can affect multiple components of HDL worsening the loss of cardioprotective effects. In summary, several studies have shown that inflammation in different pathologies can modulate LCAT activity, which in turn influences HDL levels, especially functionality. Although this observation of HDL modification by LCAT as a consequence of disease is well established, the potential causal role of LCAT-modified HDL in disease development remains rather unexplored.

### 3.3. Cholesterol Ester Transfer Protein

CETP is an enzyme which mediates the transfer of cholesterol esters from HDL to apoB-containing lipoproteins in exchange for triglycerides [83]. It has been shown in a hamster model, which endotoxin and pro-inflammatory cytokines, such as TNF-α and IL-1 decrease serum CETP activity, as well as CETP mRNA levels and protein expression in several tissues such as heart, muscle and adipose tissue [84]. These effects result in increased circulating HDL levels, which is beneficial for the clearance of endotoxins. It has also been confirmed that lipopolysaccharide (LPS) mediates a profound reduction in hepatic *CETP* gene expression in human CETP transgenic mice. Interestingly, it has been shown that these effects were primarily a result of adrenal corticosteroid release [85]. This decreased CETP activity upon inflammation has also been confirmed in cardiac surgery patients [86], reflecting a model of sterile inflammation, and in sepsis patients [81,87]. Furthermore, a decline in plasma CETP activity was also observed in patients with RA, suggesting low CETP levels might be considered an increased risk of cardiovascular mortality in RA [88]. These changes in CETP activity upon inflammation likely impact on the remodeling of HDL, thereby influencing HDL composition and function, although further studies are needed to validate this.

### 3.4. Apolipoprotein M

Apolipoprotein M (apoM) is a plasma protein of the apolipoprotein family, which is expressed in the liver and kidney [89]. It is predominantly enriched in HDL, but is also present in small quantities in LDL and VLDL [90]. ApoM has a hydrophobic binding pocket, which facilitates its binding to its natural ligands, such as retinoic acids [91] and sphingosine 1-phosphate (S1P) [92]. The concentration of apoM in the plasma is approximately 0.9 µM and ∼5% of HDL in circulation carries apoM and S1P [93,94]. The expression of apoM in the liver and kidney is decreased by inflammatory stimuli, such as lipopolysaccharide (LPS), thereby lowering the circulating levels of apoM (Figure 3) [95]. As apoM has anti-atherogenic effects, which are at least partly mediated by its close interaction partner S1P [96], these reduced apoM levels increase CVD risk. It has also been reported that the conversion of HDL to pre-beta HDL is impaired in apoM deficient mice, which leads to markedly reduced cholesterol efflux [97]. In other inflammatory diseases, such as sepsis, psoriasis and systemic lupus erythematosus (SLE), apoM levels are also reduced, which might contribute to the disturbance of HDL composition and its function [98,99,100]. Additionally, diabetes has been shown to influence apoM levels as shown by decreased S1P and apoM plasma levels in type 2 diabetic patients [101,102]. Although patients with type 1 diabetes have normal plasma apoM levels, the apoM/S1P complex shifts from small, dense HDL to larger, less dense HDL particles [103]. The association of the apoM/S1P complex with larger, less dense HDL particles attenuates the anti-inflammatory effects of HDL [103], which could lead to elevated cardiovascular disease risk associated with type 1 diabetes. All in all, it is clear that HDL-bound apoM is affected by inflammation and in various pathological conditions. However, it remains to be further validated whether these changes also contribute or exacerbate disease development.

### 3.5. Sphingosine-1-Phosphate

S1P is a lysosphingolipid found in association with small and dense HDL particles [104] consistent with higher content of apoM, a well-known carrier of S1P in small and dense particles [22,105]. It is a bioactive lipid which acts as a lipid mediator and also plays a role as a signaling molecule [106]. When associated with HDL, S1P has been found to inhibit various metabolic, inflammatory and vascular effects [107]. Moreover, studies have demonstrated that HDL-bound S1P is important for endothelial survival, angiogenesis, migration, nitric oxide (NO) production and inhibition of inflammatory responses [108,109,110,111]. Additionally, HDL-bound S1P is thought to engage both scavenger receptor B type 1 (SR-B1) and S1P receptor to mediate important biological roles such as inhibition of endothelial injury and inflammation [107]. S1P is thought to be a crucial component mediating the antioxidant activity of HDL. It has been reported that S1P is enriched with small, dense HDL_3_ particles that potentially attenuated apoptosis and LDL oxidation in endothelial cells [104]. S1P can affect the functionality of HDL as can be observed in apoM deficient mouse, which are in turn deficient in HDL-bound S1P, due to the missing carrier, and they exhibit dysfunctional HDL as demonstrated by impaired vascular barrier function [112,113]. Furthermore, HDL from patients with CAD contains lower amounts of S1P as compared to healthy controls [114]. This reduction of HDL-S1P levels can be caused by oxidative modifications, such as those known to occur in CAD patients [115]. Meanwhile, HDL in patients with acute myocardial infarction and high acute phase response also exhibit a limited ability to stimulate endothelial cells to produce NO, while presenting reduced levels of S1P [116]. Lower levels of HDL-bound S1P have also been found in other pathologies including diabetes mellitus, chronic kidney disease, and atherosclerosis, and have been linked to the impaired functionality of HDL (Figure 3) [102,117]. All of these studies clearly show that various pathologies influence HDL-bound S1P as a pathological consequence. Recently, a study by Brulhart et al. demonstrated that HDL-bound S1P can also have a causal effect in disease development as they could show that supplementing reconstituted HDL with S1P enhanced its protective capacity in an ischemic reperfusion injury mouse model [118]. This data clearly demonstrates that S1P plays an important biological role and confers important protective effects of HDL, which can be diminished following modifications in multiple human disorders that are commonly associated with inflammation.

### 3.6. Paraoxonase 1

PONs are a family of enzymes present in mammals composed of three different enzymes, which possess antioxidative properties. PON1 and PON3 are both synthesized by the liver and are associated with HDL [119]. PON1 gained significant interest as a protein that is responsible for the antioxidant properties of HDL. HDL associated PON1 possesses anti-atherogenic properties, protects LDL from oxidative modification and promotes cholesterol efflux [120]. It has been demonstrated that in obese subjects the increase in oxidative stress in HDL is associated with reduced HDL–PON activity. Lower PON activity and changes in HDL and LDL composition might contribute to greater risk of CVD in obese subjects [121,122], although a causal relationship remains to be validated. Similarly, in type 1 and type 2 diabetes serum PON1 activity was also significantly reduced [123,124,125,126]. Correspondingly, plasma PON levels are also found to be decreased in patients with inflammatory diseases such as RA, systemic lupus erythematosus (SLE), psoriasis, and infections, clearly demonstrating that these effects are observed in a broad spectrum of diseases [127,128,129,130,131,132]. One study revealed that patients with high HDL levels and low PON1 levels were more vulnerable to CHD in comparison to patients with low HDL and high PON1 [133], further highlighting the importance of PON1 activity in HDL. Additionally, PON1 has been shown to prevent the modification of other proteins through its ability to hydrolyze homocysteine thiolactone, which can contribute to atherogenesis by damaging endothelial and vascular smooth muscle cells, as well as other oxidized intermediates [134]. Overall, it is clear that reduced PON1 levels in these inflammatory pathologies result in a diminished potential for impedance of oxidation. Together, these studies demonstrate that inflammation can influence HDL-associated PON1 (Figure 4) and, thereby, indirectly also HDL function. Future research should focus on the elucidation of the potential causal role that modifications in HDL-bound PON1 can have in disease development as above described studies only demonstrate a consequential role or association at best.

### 3.7. Myeloperoxidase

As previously mentioned, MPO, a leukocyte-derived heme protein, is another protein that is involved in modifications to HDL. MPO binds to HDL by selectively targeting apoA-1 [135,136]. As a source of reactive oxygen species (ROS), MPO oxidizes apoA-1 (Figure 5), thus, negating the atheroprotective effects that HDL exerts by impairing the ability of HDL to participate in cholesterol efflux [137]. This effect is mainly caused by the inability of oxidized apoA-1 to be separated from HDL into a lipid-free or lipid-poor form [138]. This separation is a critical step in cholesterol efflux, as lipid-poor and lipid-free apoA-1 is the main acceptor of cholesterol and phospholipids from macrophages [139]. LCAT activity is also diminished as its activating protein, apoA-1, is compromised [135,136], which has its own set of effects as discussed previously. Moreover, studies have demonstrated a link between the ROS generated by MPO and atherogenesis, suggesting that the MPO oxidative pathway is a major participant in HDL modification [136]. These proatherogenic and pro-inflammatory effects are evident in HDL and apoA-1 recovered from human atherosclerotic lesions, which are dysfunctional and highly oxidized by MPO [140]. High MPO activity has been identified in various disorders, such as RA and SLE, and are accompanied by a reduction in the antioxidant effects exhibited by HDL. HDL from patients suffering from these pathologies also demonstrated an impaired cholesterol efflux capacity [22]. Huang et al. were able to show that MPO and PON1 modulate the function of one another in vivo as demonstrated in both a mouse model of inflammation and in human clinical studies of subjects exhibiting acute inflammation. As previously mentioned, PON1 exerts systemic antioxidant effects as a complex with HDL while MPO is a source of oxidation. In their study, Huang et al. describe a ternary complex in HDL where PON1 can partially inhibit MPO, and MPO can inactivate PON1. Their research concluded that oxidation catalyzed by MPO resulted in modification and inactivation of PON1 (Figure 5) both in vitro and in vivo. Due to the protective role of PON1 and its association with HDL, they propose that MPO generated modification of PON1 is one of the mechanisms by which dysfunctional HDL forms in response to inflammation [137]. MPO contributes to the pathogenesis of CVD through its ability to modify key proteins in HDL, and as evidence has shown, it has a clear negative impact on the protective properties of HDL [22]. Although it is clear that in various pathologies MPO modifies HDL, more research is required to determine whether MPO modified HDL can also play a causal role in disease development.

## 4. Targeting Dysfunctional HDL: Therapeutic Approaches

Dyslipidemia, the lipid disorder that accompanies many of the mentioned disorders, such as atherosclerosis, is characterized by elevated triglycerides, LDL-C, and reduced levels of HDL-C [141,142]. Therefore, the classical approach to normalize these lipids has aimed to decrease LDL-C and other atherogenic lipoproteins by administering statins and to increase HDL-C [143]. For example, a direct approach consisted of HDL infusions, which increase levels of HDL-C and decrease plaque and inflammation markers. However, so far almost all studies aimed at raising HDL-C have failed to demonstrate beneficial effects on disease risk or mortality [144]. In a comprehensive review, many more therapies, ranging from regulatory microRNA to pharmacological treatments with niacin, CETP inhibitors, fenofibrates, which raise HDL levels are reviewed [26]. However, as revealed in an exhaustive review from Nazir et al., targeting composition and functionality of HDL, rather than HDL-C, has a much higher clinical potential [12]. In this review, we have also meticulously described the clinical potential of various modifications of HDL in a wide range of pathologies, such as diabetes, auto-immune diseases, and chronic kidney disease [12].

As can be inferred by the plethora of proteins and lipids that make up HDL, there is a multiplicity of potential targets that can be used for potential therapies to improve HDL functionality. Here, we will highlight some of the strategies that are being employed or investigated in order to improve HDL function and reduce CVD risk [145]. It still needs to be further validated whether (and by which mechanisms) dysfunctional HDL plays a causal role in disease development. Nevertheless, several approaches are already being investigated to improve HDL function. The use of reconstituted HDL has been suggested to improve the functionality of HDL [115], as its composition can be altered to one that has improved functionality, potentially by changing the concentration of some components such as apoA-1 or S1P [118]. For example, autologous plasma infusions after delipidation of HDL was tested in patients with acute coronary syndrome (ACS). Although the infusions were a well-tolerated and feasible treatment, it only showed a mild and not significant reduction of atheroma volume in comparison to controls [146]. Meanwhile, the ApoA-I Event Reducing in Ischemic syndromes I (AEGIS-I) trial used infusions of CSL112, a reconstituted plasma-derived apoA-1, in patients with acute myocardial infarction. These infusions were also tolerated well and confirmed to enhance cholesterol efflux [147]. Although showing promising initial results, further assessment is required for both CSL112 and the delipidated HDL infusions. Additional results regarding CSL112 are expected in June 2022 through the AEGIS-II Phase III clinical trial focused on evaluating the safety and efficacy of CSL112 in the reduction of risk of adverse cardiovascular events in a large cohort of patients with ACS [148]. Another approach focused on S1P to improve the anti-inflammatory function of HDL [117]. In this in vitro study, S1P was incorporated into HDL derived from both wild-type and S1P deficient mice, in which the enhanced HDL from wild-type mice showed improved capacity to inhibit TNF-α induced gene expression [116]. Moreover, HDL enriched with S1P from S1P deficient mice, which previously lacked any inhibitory effects, demonstrated potent inhibition of inflammation. Additionally, similar improvement in inhibitory effects were observed with S1P-loading of healthy human HDL and HDL from CAD patients [116]. ApoA-1 mimetic peptides are also the subject of investigation for improvement of HDL functionality, as various studies have found that these peptides can improve some of the functional properties of HDL [149], for example, increasing the HDL-mediated cholesterol efflux, increasing PON1 activity, and reducing atherosclerosis as reviewed by Kontush et al. [115]. For example, D-4F, an orally active peptide, was tested in cynomolgus monkeys, which have pro-inflammatory HDL, and was shown to be highly efficient in reducing the index of inflammation within 2 hours from administration. Additionally, HDL derived from D-4F treated monkeys was shown to have an increased cholesterol efflux capacity from human macrophages [150]. Reduction in the inflammatory index was also observed in preliminary data obtained from patients with CAD treated with D-4F as opposed to placebo treated controls [17]. Thus, it can be inferred that D-4F affects the composition and functionality of HDL. Inhibition of MPO has also been proposed as a treatment due to its deleterious effects on HDL and its main components. Several inhibitors have shown some promising results, while exhibiting different protective effects. For example, PF-1355 was found to improve cardiac function and prevent cardiomyopathy due to ischemia [151]. These effects could be at least partly linked to reduced MPO-driven modifications and thereby minimized dysfunction of HDL. Although MPO inhibitors demonstrate some promising effects, further evaluation is required as MPO plays an essential role in the protection against microbes [152] and, thus, MPO inhibition may negatively impact a necessary immune response [153]. As the interplay between the immune response and HDL is so complex, the challenge of producing purely beneficial effects with the proposed therapies is immense. Overall, there are a myriad of potential targets and only a very limited amount is currently being investigated. Improved understanding of the modifications to the lipidome and proteome of HDL and their effects on HDL function could lead to novel and better focused therapies. Furthermore, investigations focusing on the effect of dysfunctional HDL on associated diseases and its potential to result in the progression of the pathology will also be highly valuable to elucidate more suitable therapeutic strategies. 

## 5. Concluding Remarks

HDL is very heterogeneous as it consists of various subclasses and is subject to constant dynamic remodeling through the interaction with plasma constituents, tissues, and cells. At the same time, HDL is subjected to a plethora of pathophysiological modifications in cardiovascular and metabolic diseases. In this review the role of HDL in various metabolic and inflammatory diseases has been discussed, focusing especially on HDL composition and function together with an overview of potential therapeutic targets. Previously it was thought that raising HDL-C levels was a promising way to reduce the risk of cardiovascular mortality. However, increasing evidence indicates that the composition and function of HDL is a more important determinant for disease outcome than HDL-C levels. The molecular mechanisms of dysfunctional HDL are still insufficiently understood, but generally thought to be modifications (such as oxidation and glycation) in HDL lipidome and proteome that affect the composition of HDL and, thereby, its functionality. So far, most studies have focused on the effects of various pathologies on HDL composition and function, though the potential causal affects that such modifications can have on disease development are still rather unexplored. Therefore, future research should focus on the elucidation of these causal pathogenic effects of HDL modifications. Furthermore, extensive research is needed to identify new agents with favorable side effects to improve HDL composition and functionality and potentially reduce disease risk.

## Figures and Tables

**Figure 1 biomedicines-08-00549-f001:**
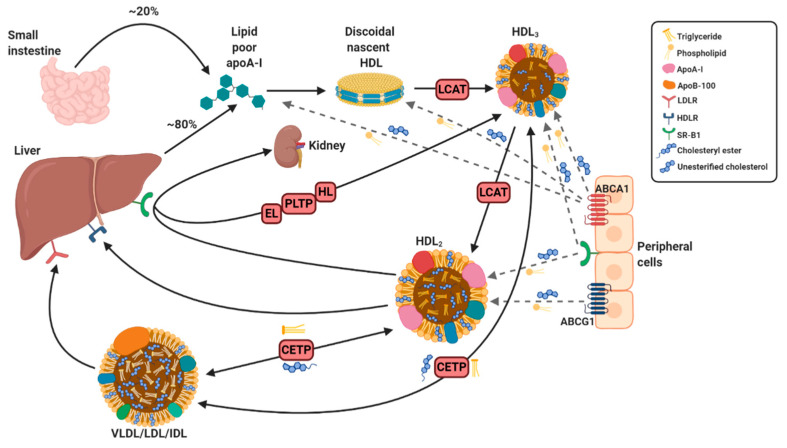
Simplified overview of HDL synthesis and metabolism. The liver and intestines produce apoA-I which becomes discoidal nascent HDL after interaction with ATP-binding cassette transporter A1 (ABCA1). Lipidation of discoidal nascent HDL with unesterified cholesterol and phospholipids via ABCA1, ABCG1, and scavenger receptor B type 1 (SR-B1) and cholesterol esterification via lecithin-cholesterol acyltransferase (LCAT) produces spherical HDL_3_ and further lipidation results in HDL_2_ (only representative cross sections shown). Cholesterol is transported to the liver directly from HDL_2_ via a not yet identified HDL receptor (HDLR) or through binding with SR-B1 (to be excreted by kidneys) or indirectly via interaction of very-low-density lipoprotein (VLDL), low-density lipoprotein (LDL), and intermediate-density lipoprotein (IDL) with LDL receptor (LDLR) following uptake of cholesterol and triglycerides from HDL_2_, which is facilitated by cholesterol ester transfer protein (CETP). Recycling of HDL_2_ occurs by the remodeling into lipid-poor and denser HDL_3_ catalyzed by endothelial lipase (EL), hepatic lipase (HL), and phospholipid transfer protein (PLTP), which can once again undergo further lipidation. Solid arrows refer to specific steps in HDL metabolism and maturation, while dotted arrows reflect the transfer of triglycerides and unesterified cholesterol from peripheral cells to specific particles by interaction with displayed receptors.

**Figure 2 biomedicines-08-00549-f002:**
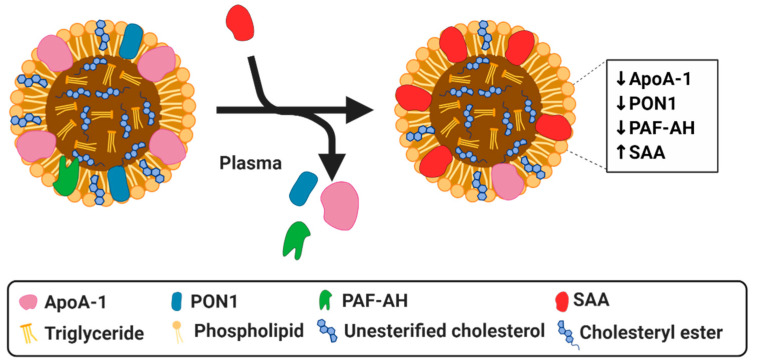
Simplified diagram of serum amyloid A (SAA) modified HDL particle resulting in a dysfunctional HDL. SAA can displace apoA-I, PON1 and PAF-AH.

**Figure 3 biomedicines-08-00549-f003:**
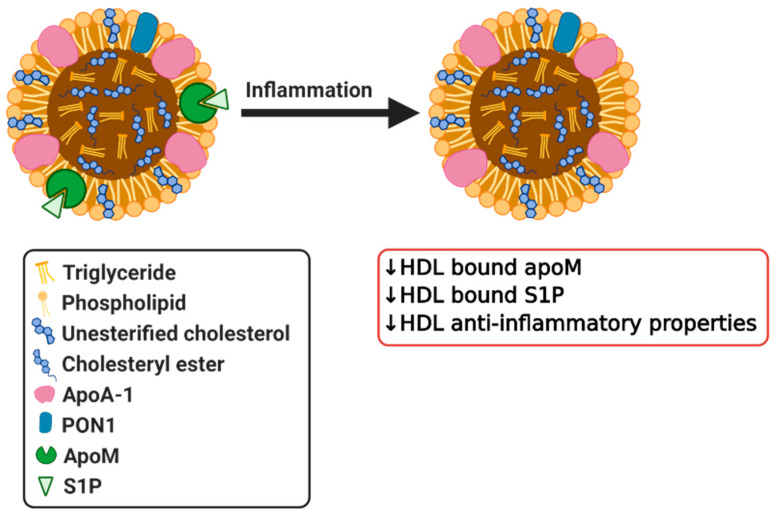
Simplified diagram of effects of inflammation on HDL bound apoM and sphingosine 1-phosphate (S1P) resulting in dysfunctional HDL.

**Figure 4 biomedicines-08-00549-f004:**
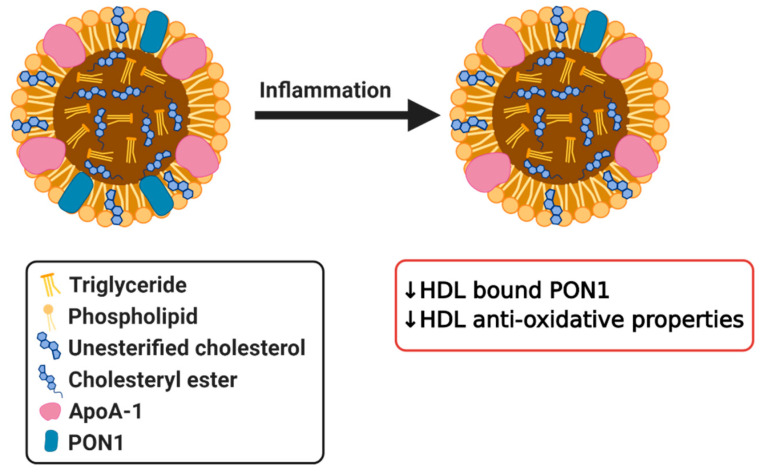
Simplified diagram of effects of inflammation on HDL bound paraoxonase 1 (PON1) resulting in dysfunctional HDL.

**Figure 5 biomedicines-08-00549-f005:**
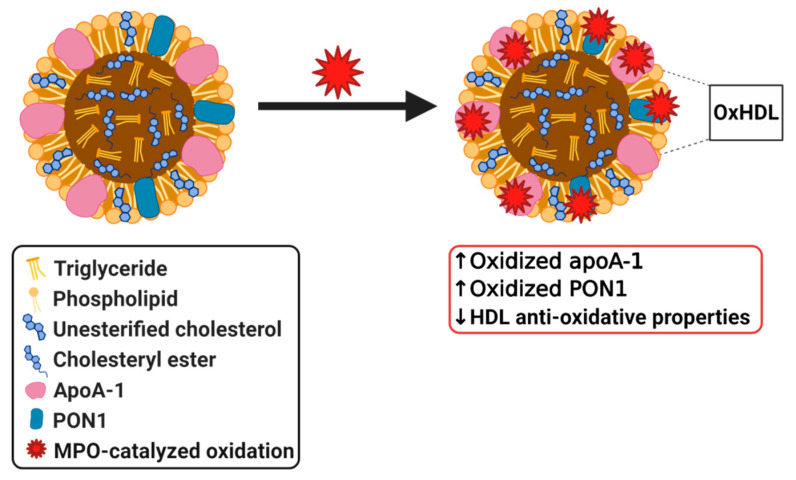
Simplified diagram of myeloperoxidase (MPO) modified dysfunctional HDL.

**Table 1 biomedicines-08-00549-t001:** Major high-density lipoprotein (HDL) subclasses according to different separation techniques.

Ultracentrifugation	Gradient Gel Electrophoresis	2D Electrophoresis	Immunoaffinity Chromatography
HDL_2_ (density 1.063–1.125 g/mL)	HDL_2b_ (particle size 10.6 nm)	Pre-α HDL particles	HDL with only apoA-I
HDL_3_ (density 1.125–1.21 g/mL)	HDL_2a_ (particle size 9.2 nm)	Pre-β HDL particles	HDL with only apoA-II
	HDL_3a_ (particle size 8.4 nm)		HDL with apoA-I and apoA-II
	HDL_3b_ (particle size 8.0 nm)		
	HDL_3c_ (particle size 7.6 nm)

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
