# Peer review of "High-Density Lipoprotein Modifications: A Pathological Consequence or Cause of Disease Progression?"

_biomedicines, 2020, doi:10.3390/biomedicines8120549_

Round 1

Reviewer 1 Report

This is a nice and straightforward review.

I would recommend the development of some tables and more figures to accompany and easen the text readability.

Well this is up to date review of the topic which is a very wide one. It is well done and off course given the fact that is a review there is no novelty besides the fact that pit provides the current status of evidence of the topic. As I stated in the review of the manuscript, this is lacking some non textual content which can significantly improve readability for professionals not very specialized on the topic. My recommendation is that it is a good manuscript and should be published but with a addition of figures and table.

Author Response

I would recommend the development of some tables and more figures to accompany and easen the text readability.

Well this is up to date review of the topic which is a very wide one. It is well done and off course given the fact that is a review there is no novelty besides the fact that pit provides the current status of evidence of the topic. As I stated in the review of the manuscript, this is lacking some non-textual content which can significantly improve readability for professionals not very specialized on the topic. My recommendation is that it is a good manuscript and should be published but with an addition of figures and table.

We would like to thank the reviewer for the insightful review of our manuscript and we appreciate the positive evaluation and the constructive suggestions that he/she raised. We have included more figures and a table according to the reviewer’s suggestions.

Reviewer 2 Report

This is a well written, interesting review that summarized an interesting target for drug development. All the information presented is correct, however, some relevant information is missing and its incorporation into the manuscript would strengthen it.

The abstract should be more informative and attractive since the manuscript is a review. It would be advisable to state in the abstract the actual changes in HDL composition that could lead to pathology and in a stronger way indicate that to modulate the HDL composition probably is a molecular target to treat diseases. Since abstract in the actual form is short these changes could be easily accommodated on it.

A figure indicating the synthesis of HDL at the liver, the processing, and its role in peripheral tissues would be also an addition to the manuscript. More figures would make the article more attractive.

Along with the manuscript, modifications that take place in the HDL are described, for example, the substitution of Apo-1 by SAA, and the effects that these substitutions have been described, however, the molecular bases of why these proteins are changed could be presented to highly the molecular bases of the pathology.

In the introduction, changes in lipid composition, lipidomics, glycation, oxidative stress, and so on are just mentioned. A chapter to develop the information on these changes, mainly in diabetes would be also advisable. In the present state only changes in protein composition are presented.

In the therapeutic approach, a reorganization using the different diseases as a leading guide would clarify the information presented.

Author Response

Reviewer #2

  1. This is a well written, interesting review that summarized an interesting target for drug development. All the information presented is correct, however, some relevant information is missing and its incorporation into the manuscript would strengthen it.

The abstract should be more informative and attractive since the manuscript is a review. It would be advisable to state in the abstract the actual changes in HDL composition that could lead to pathology and in a stronger way indicate that to modulate the HDL composition probably is a molecular target to treat diseases. Since abstract in the actual form is short these changes could be easily accommodated on it.

We thank the reviewer for the thorough and positive evaluation of our manuscript. As suggested, we have added additional information in the abstract about the modifications discussed in the review and commented on their potential as therapeutic targets keeping within the word count limitations prescribed by the journal.

  1. A figure indicating the synthesis of HDL at the liver, the processing, and its role in peripheral tissues would be also an addition to the manuscript. More figures would make the article more attractive.

As suggested by the reviewer, we have included a figure focusing on the HDL metabolism (Figure 1 on page 4 of the revised manuscript). Additionally, more figures focusing on specific modifications are now included in the manuscript.  

  1. Along with the manuscript, modifications that take place in the HDL are described, for example, the substitution of Apo-1 by SAA, and the effects that these substitutions have been described, however, the molecular bases of why these proteins are changed could be presented to highly the molecular bases of the pathology.

Additional information on the affinity of SAA to HDL has been incorporated, including the reasoning for its ability to displace apoA-1 as suggested by the reviewer (Line 46-47 on page 5 and Line 1-3 on page 6 of the revised manuscript).

  1. In the introduction, changes in lipid composition, lipidomics, glycation, oxidative stress, and so on are just mentioned. A chapter to develop the information on these changes, mainly in diabetes would be also advisable. In the present state only changes in protein composition are presented.

We appreciate this comment raising an important point. We have included a separate chapter in revised manuscript focusing on the post-translational modifications of HDL providing more information in addition to changes in protein composition. In this chapter, also modifications in a diabetic situation are discussed/highlighted (Chapter 2, starting on page 4 of the revised manuscript).

  1. In the therapeutic approach, a reorganization using the different diseases as a leading guide would clarify the information presented.

Although we appreciate the suggestion by the reviewer, we believe that a reorganization is not appropriate/required based on the main focus of the current review. In this review we primarily focus on some the general modifications of HDL, which are rather similar in various diseased states, and therefore discussed together. However, recently we have published an elaborate review that specifically focusses on HDL in various pathologies. We have added a comment to reference to this recent review in chapter 4 (Line 13-15 on page 11 of the revised manuscript) to direct interested readers to this manuscript focusing on disease-specific observations.

Round 2

Reviewer 2 Report

All changes incorporated in the manuscript have improved the readability. Figures, new chapters and improved abstract make the review more attractive.